# Proton Therapy for Squamous Cell Carcinoma of the Head and Neck: Early Clinical Experience and Current Challenges [note 1]

**DOI:** 10.3390/cancers14112587

**Published:** 2022-05-24

**Authors:** Sandra Nuyts, Heleen Bollen, Sweet Ping Ng, June Corry, Avraham Eisbruch, William M Mendenhall, Robert Smee, Primoz Strojan, Wai Tong Ng, Alfio Ferlito

**Affiliations:** 1Laboratory of Experimental Radiotherapy, Department of Oncology, Katholieke Universiteit Leuven, 3000 Leuven, Belgium; heleen.bollen@uzleuven.be; 2Department of Oncology, Leuven Cancer Institute, Universitair Ziekenhuis Leuven, 3000 Leuven, Belgium; 3Department of Radiation Oncology, Austin Health, The University of Melbourne, Melbourne, VIC 3000, Australia; sweetping.ng@austin.org.au; 4Division of Medicine, Department of Radiation Oncology, St. Vincent’s Hospital, The University of Melbourne, Melbourne, VIC 3000, Australia; june.corry@genesiscare.com; 5Department of Radiation Oncology, University of Michigan, Ann Arbor, MI 48109, USA; eisbruch@med.umich.edu; 6Department of Radiation Oncology, College of Medicine, University of Florida, Gainesville, FL 32209, USA; mendwm@shands.ufl.edu; 7Department of Radiation Oncology, The Prince of Wales Cancer Centre, Sydney, NSW 2031, Australia; robert.smee1@health.nsw.gov.au; 8Department of Radiation Oncology, Institute of Oncology, University of Ljubljana, 1000 Ljubljana, Slovenia; pstrojan@onko-i.si; 9Department of Clinical Oncology, Li Ka Shing Faculty of Medicine, The University of Hong Kong, Hong Kong, China; ngwt1@hku.hk; 10Coordinator of the International Head and Neck Scientific Group, 35125 Padua, Italy; profalfioferlito@gmail.com

**Keywords:** proton therapy, head and neck cancer, early experience

## Abstract

**Simple Summary:**

Proton therapy is a promising type of radiation therapy used to destroy tumor cells. It has the potential to further improve the outcomes for patients with head and neck cancer since it allows to minimize the radiation dose to vital structures around the tumor, leading to less toxicity. This paper describes the current experience worldwide with proton therapy in head and neck cancer.

**Abstract:**

Proton therapy (PT) is a promising development in radiation oncology, with the potential to further improve outcomes for patients with squamous cell carcinoma of the head and neck (HNSCC). By utilizing the finite range of protons, healthy tissue can be spared from beam exit doses that would otherwise be irradiated with photon-based treatments. Current evidence on PT for HNSCC is limited to comparative dosimetric analyses and retrospective single-institution series. As a consequence, the recognized indications for the reimbursement of PT remain scarce in most countries. Nevertheless, approximately 100 PT centers are in operation worldwide, and initial experiences for HNSCC are being reported. This review aims to summarize the results of the early clinical experience with PT for HNSCC and the challenges that are currently faced.

## 1. Introduction

Head and neck squamous cell carcinoma (HNSCC) is the seventh most common cancer and cause of cancer-related death worldwide, with 878,000 new cases and 444,000 deaths yearly [1]. Usually, HNSCC is diagnosed at a locally advanced but curable stage, for which radiotherapy (RT), with or without concomitant radio-sensitizing chemotherapy, is the recommended first-line treatment in over 80% of cases [2]. The delivery of high-dose RT (typically at around 70 Gray) to the gross tumor, i.e., the ‘gross target volume’ (GTV), is necessary to achieve cure [3,4]. However, the planning and delivery of RT in patients with HNSCC is complex, due to the close proximity of the tumor to surrounding critical organs at risk (OAR). Over the last decade, the implementation of intensity-modulated radiotherapy (IMRT) has led to higher conformity of radiation dose distribution, with superior coverage of the target volumes and a reduced dose to critical OARs, compared to older photon techniques [5]. However, dose to the surrounding OAR cannot be completely avoided, and is associated with significant acute and long-term toxicities in patients with HNSCC [6]. The recent epidemiologic shift towards HPV-driven disease in oropharynx cancer (OPC), with improved prognosis compared to non-HPV-associated counterparts, has led to a significant number of long-term survivors who may be affected by late treatment-related toxicities, such as xerostomia, dysphagia, neurologic complications, and tube feeding dependence, for which limited treatment options exist. [7,8]. Protons, that is, positively charged particles, deposit energy in tissue at a certain depth by virtue of a phenomenon called the Bragg peak (Figure 1).

As the energy deposition takes place within a narrow range of depth in tissue, PT offers the ability to deliver high doses to the TV while minimizing the dose to the adjacent OAR, compared to photon therapy [9]. Preclinical and dosimetric studies comparing PT to IMRT have consistently demonstrated significant reductions of OAR dose, suggesting that PT has the potential to decrease acute and long-term toxicities [10,11,12,13,14]. Van de Water et al., reported the dose for all OAR to be reduced in nasopharyngeal cancer (NPC), while for OPC, hypopharyngeal (HPC), and laryngeal cancer (LC) the dose to the parotid glands, larynx, and spinal cord was reduced [14,15]. Similarly, when comparing dosimetry between intensity-modulated proton therapy (IMPT) and IMRT treatment plans for NPC, Lewis et al., reported a significant reduction in the mean dose for 13 out of 29 OAR, including oral cavity, brain, spinal cord, and larynx-favoring IMPT plans [16]. A major criticism to existing dosimetric studies for PT is that setup errors and anatomical changes were not taken into account. Furthermore, is remains unclear whether dose reduction to OAR provides a significant clinical impact when dose levels of both the PT and IMRT treatment plan are within the defined tolerance. Since the observed differences may be overestimated, clinical evaluation is necessary. Clinical evidence for PT in patients with HNSCC is limited, and level I evidence for PT is lacking [17]. Therefore, it has been a challenge to identify patients that would benefit the most from PT. As a consequence, established indications for the reimbursement of PT remain scarce in most countries. Nevertheless, approximately 100 PT centers with over 230 treatment rooms are in operation worldwide, and initial experiences with PT for HNSCC are being reported [18]. This review aims to summarize early clinical experience with PT for patients with HNSCC, and the current challenges for further implementation.

## 2. Early Clinical Experience

To date, the clinical experience with PT for HNSCC in a definitive setting is limited. Most studies evaluating PT for head and neck cancer (HNC) either focus on non-SCC histology or take place in the re-irradiation setting, in which the clinical outcomes are expected to be different compared to the definitive setting [19,20,21,22]. Furthermore, several groups have published early outcomes of a mixed-beam (MB) approach of photons combined with PT for HNSCC [23,24].

To date, however, series reporting clinical outcomes after single-modality PT for HNSCC in a curative setting remain limited. As the number of institutions practicing PT increases, more positive results, albeit based on small studies, are being reported. Table 1 provides an overview of all series evaluating both toxicity and oncologic outcomes after single-modality PT for HNSCC. Studies investigating other heavy ion particles, such as carbon ions, non-SCC etiologies, and multiple HNC locations were not included.

### 2.1. Proton Therapy in the Definitive Setting

Theoretically, SCC of the nasopharynx (NPSCC) is among the most accepted indications for PT, given the complex anatomy and proximity to important OARs, such as the chiasm and optic nerve, brainstem, temporal lobes, pharyngeal constrictor muscles, and major salivary glands. Approximately 50 to 75% of patients with NPC treated with IMRT develop acute grade 3 or 4 toxicities, with 10 to 20% of surviving patients reporting serious late effects [38,39]. Six series reporting their early clinical experience with PT for NPSCC, as summarized in Table 1. With a median follow-up time of 27 months, all report a reduction of treatment-related toxicities compared to IMRT, acceptable compliance with combined chemotherapy and PT, and excellent oncologic outcomes. A significant reduction of acute grade 2 or higher toxicities compared to IMRT in NPSCC patients was shown by Li et al., and Holliday et al., [25,34]. Several studies reported reactive feeding tube placement rates to be lower in patients with NPSCC treated with PT, compared to those treated with IMRT [25,37]. Holliday et al., reported that the lower feeding tube rate was strongly associated with a reduction in the mean oral cavity dose. A reduction in tube feeding placement was also demonstrated in patients with SCC of the oropharynx (OPSCC) [26,28,31]. Of note, the rate of feeding tube placement varies greatly among studies, probably reflecting a different practice preference in the management of acute adverse events (AE) during treatment. A significant decreases in oral cavity dose using PT has been reported reproducibly, probably not only affecting the rate of tube feeding placement, but also leading to reduced xerostomia and dysgeusia [36,40]. Zhang et al., found a significant reduction in the mean mandibular dose and percent volumes (V45–V70) in 50 patients with OPSCC treated with PT, compared to IMRT (25.6 Gy vs. 41.2 Gy), translating to a decreased incidence in osteoradionecrosis [29]. In the largest retrospective study of 532 patients with OPSCC (429 PT and 103 IMRT), the proportion of moderate-to-severe xerostomia was significantly lower (14%) in the PT group at 18–24 months and 24–36 months [36]. These findings indicate that PT might be a valuable treatment option to achieve a reduction in long-term treatment-related toxicities, which is especially interesting considering the increasing number of young patients with good prognosis HPV-related OPSCC.

Overall, PT appears to be an interesting RT modality with excellent treatment outcomes in patients with HNSCC. However, as a consequence of a spread-out Bragg peak increasing the entrance skin dose, PT inherits an increased risk for radiation-induced dermatitis (RD), which has been extensively demonstrated in breast cancer patients with PT [41]. In a propensity score-matched analysis of 160 patients with NPC, Chou et al., reported that RD was a significant acute side effect in the PT group (35% compared to 7.5% in the VMAT group) [37]. Lewis et al., also reported that 40% of 10 patients treated with PT to have developed grade 3 RD [16], while Aljabab et al., found 76% to experience grade 3 RD in a cohort study of 46 patients [30]. Similarly, Romesser et al., reported a significant difference between PT and photon therapy, with 100% of patients treated with PT experiencing grade 3 RD [42]. A trend towards increased RD in the PT group was also reported by Manzar et al. [31], while Blanchard et al., reported no difference for grade 3 or higher RD [28]. The variability in the incidence of RD between series presumably reflects variable experience with PT planning among institutions. Therefore, the incidence of RD may decrease with increasing experience in PT planning. Moreover, one could consider that skin toxicity is a price we are willing to pay in exchange for reduced AEs with higher impact on quality of life (QOL).

Table 1 is limited to studies with NP and OPSCC patients. To the best of our knowledge, no clinical studies regarding PT for laryngeal/hypopharyngeal SCC exist. Only one abstract was published on the outcomes and toxicities of 14 patients treated with PT for laryngeal SCC. The authors report a high LRC and OS with a cumulative incidence rate of 12.3% and 8.3% for acute and late grade 3 toxicities, respectively [43]. The nasal cavity and paranasal sinuses represent an anatomic site well-suited for the application of PT. However, these malignancies are rare and composed of several histological types, rendering comparison with IMRT difficult. A large meta-analysis of 41 observational studies found PT to be associated with significantly higher loco-regional control, compared to IMRT for nasal cavity and paranasal sinus malignancies, at a follow-up of 38 months [44]. Of note, only 27% of patients treated with PT had a high-risk histological subtype, including SCC, but also undifferentiated, poorly differentiated carcinoma, possibly influencing the difference in oncologic outcomes. No difference in toxicity was reported between the 2 groups, aside from a higher rate of neurological complications in the PT group (20% of patients, compared to 4% in the photon group). However, there may have been reporting bias with a larger proportion of PT studies that reported treatment-related toxicities [44]. In a series by McDonald et al., toxicity endpoints were evaluated for 26 patients treated with IMRT and 14 patients treated with PT for tumors of the nasopharynx, nasal cavity, or paranasal sinuses. Lower feeding tube dependence rate and opioid requirement were reported, likely related to lower mean dose to oral cavity and pharyngeal constrictor muscles [45]. However, similar to the series reported by Patel et al., only 3 out of 14 patients treated with PT had SCC histology.

### 2.2. Proton Therapy for Dose (De-)Escalation, Adjuvant Treatment and Re-Irradiation

There has been growing interest in treatment de-escalation by reducing the volume of the electively irradiated neck for well-selected patients with lateralized oral cavity and oropharyngeal SCC [46]. Several studies have investigated the benefits of PT for unilateral neck irradiation for parotid gland tumors and skin cancers, reporting excellent organ sparing and considerably lower side effects compared to historical IMRT outcomes [42,47,48,49]. However, the evidence for unilateral PT in HNSCC remains rather limited to date. In a recent publication by the Mayo Clinic, dosimetry and patient-reported outcomes (PROs) were compared for 40 patients undergoing ipsilateral neck irradiation with IMPT or VMAT for tonsil or salivary gland cancer. Of the patients treated with IMPT, only five were treated for (tonsillar) SCC. The authors report a significant lower mean dose on the pharyngeal constrictor muscles, contralateral parotid glands, larynx, and oral cavity, resulting in less xerostomia, dysphagia, and feeding tube placement [50]. The phase II SAVER trial (Clinicaltrials.gov: NCT04609280) aims to investigate volume de-escalation in HPV-related OPSCC with either IMRT or PT, with 2-year out-of-field contralateral nodal failure as primary endpoint.

Alongside de-escalation strategies, PT may allow for dose escalation while sparing non-target structures. Alterio et al., reported a series of 27 patients with locally-advanced NPC, treated with a MB approach of IMRT to 54–60 Gy, followed by an IMPT boost to 70–74 Gy relative biological effectiveness (RBE), to the GTV. Compared with a historic cohort of NPC patients treated with IMRT alone to 69.96 Gy, the MB approach was associated with a reduction in grade 3 mucositis [23]. Slater et al., described 29 patients with stage II-IV OPSCC treated with an accelerated MB approach, to a total dose of 75.9 Gy (RBE), reporting increased loco-regional control without increased toxicity [51]. In a series of 17 NPSCC patients treated with a MB-approach with PT-boost up to a total dose of 70–78 Gy, Beddok et al., reported a 5 year local-recurrence-free survival of 86%. However, six patients developed temporal lobe necrosis [52].

There is a paucity of data regarding the benefits of PT in the postoperative setting for HNSCC. Only two series in Table 1 included some patients in the adjuvant setting [31,32]. Sharma et al., investigated patient-reported outcomes of 64 OPSCC patients who were treated with either PT of photon therapy after trans-oral robotic surgery. Several dosimetric advantages were found, reflected in higher scores in both head and neck specific and general QOL measures. Most notable was significantly less xerostomia at both 6 and 12 months after treatment in the PT group [53].

In the re-irradiation setting, PT appears to have a relatively safe toxicity profile, with acceptable outcomes compared to historical IMRT outcomes [17,54]. In the largest multi-institutional study of 92 patients re-irradiated with PT for recurrent head and neck cancer by Romesser et al., 1 year locoregional failure was 25.1%. Compared to IMRT, patients treated with PT had lower grade 3 or 4 late toxicities rates [20]. However, the reported frequency of acute and late AEs is still rather high in most studies [55,56]. Furthermore, due to the heterogeneity in cases, it is difficult to draw general conclusions about the benefits of PT for re-irradiation of HNSCC.

The major limitation of the current clinical evidence is the retrospective, single-institutional design of most series, with the small sample sizes and heterogeneous tumor primary sites. Only the series of Manzar et al., performed a sub-group analysis, revealing the most pronounced benefits in patients treated with concomitant chemoradiotherapy [31]. For the historical case-matched cohorts comparing IMRT vs. PT, PT is, of course, from a much more recent era. The median follow-up time of the series in Table 1 was only 26 months for the PT group, while some recurrences, especially for NPSCC and HPV-related OPSCC, may occur later in time. The reported benefits in toxicities for PT were generally the acute and subacute phase. Several series reporting long-term toxicities found no differences between the two modalities [26,28,31]. This is important as it could be argued that late side-effects are most relevant, both from the patient’s and society’s point of view. Interestingly, the largest and most recent reported cohort of 103 patients treated with IMPT found a significantly lower proportion of late moderate-to-severe xerostomia, compared to the IMRT group [36].

Another factor regarding the short median follow-up time is the uncertainty around the estimations for the risk of secondary tumors. Due to remaining uncertainty about the RBE at the end of the Bragg’s peak, the radiation dose to the TV might be underestimated. High-dose irradiation has been proven to be correlated with an increased risk of secondary tumor induction, with a linear dose-response curve for most organs [57,58]. In an in vivo study on mice, PT was found to cause more complex DNA damage compared to X-rays, leading to increased oxidative stress [59].

Lastly, one needs to consider the potential selection bias favoring patients who received PT, which is presumably the reason for significantly better oncologic outcomes in some trials [37]. Although these limitations complicate a direct comparison with current state-of-the-art IMRT, the early clinical experience with PT seems encouraging and worth further evaluation, both in terms of toxicity and disease control.

## 3. The Challenge of Patient Selection

The studies listed in Table 1 all include a small number of patients, limiting meaningful subset analyses to potentially guide patient selection. Given the growing availability of PT around the world, the question arises of how to optimally select patients who are likely to gain the most benefit from PT. Several RT societies have published recommendations on this matter [60,61,62], although it remains unclear to what extent they have been adopted in routine daily practice [18]. Different approaches for patient selection have been published. In 2014, the Proton Priority System was proposed by the University of Pennsylvania. The system uses a weighted sum of seven domains including diagnosis, anatomic site, stage, performance status, and comorbidities, age, urgency, and protocol participation [63]. Cheng et al., developed a prototype for an online platform for proton decision support, comparing photon and proton treatments on dosimetric, toxicity, and cost-effectiveness levels [64]. The University of Adelaide has proposed a Markov simulation framework, combining all dosimetric data to provide an estimated quality adjusted life expectancy from a given treatment plan. Their comprehensive model took into account the normal tissue complication probability (NTCP), the tumor control probability (TCP), and second primary cancer induction probability [65].

In a recent publication, Tambas et al., performed an electronic questionnaire to investigate the current practice for adult patient selection among 22 European PT centers, and found major differences in patient selection for PT. Most frequently mentioned factors for selecting patients with HNSCC for PT were as follows: (1) locally advanced HNC with primary tumor close to skull base, (2) tumors of nasopharynx, (3) unilateral neck irradiation, and 4) dose reduction to the brain. Furthermore, young age, favorable prognosis, and previous irradiation in the head and neck region were other important factors that led to the clinicians’ preference for PT [18]. It is generally accepted that PT is a preferred treatment option for pediatric HNC patients, due to the advantage of low dose spread in tissue, with a potential reduction in secondary malignancies [66].

For adults, however, various factors determine whether an individual patient will benefit from PT. Therefore, it is interesting to consider individual patient and tumor characteristics, rather than to simply rely on rigid protocols. The model-based approach represents a more individualized approach to select patients for PT. This approach was accepted as the uniform, national selection strategy in the Netherlands, and is based on the principle that the risk of radiation-induced side effects can be reliably predicted by multivariable normal tissue complication probability (NTCP) models, describing the relationship between dose-volume parameters and the risk on a certain AE of a particular grade [67]. Patients qualify for PT if the difference in dose (Δdose) based on the photon vs. PT plan comparison translates into a clinically significant ΔNTCP. This selection procedure has several advantages. Firstly, model-based selection has been proven to be more cost-effective than treating all HNC patients with either IMRT or PT [68]. Secondly, the model-based approach is an important step towards individualized cancer care. It represents a dynamic way to select patients suitable for PT, as additional complications can be easily added to the models when NTCPs become available. The challenge, however, is to timely and accurately detect possible deviations from the original NTCP-models and to adjust the modelling in a timely manner. In fact, the University of Groningen is currently working towards a comprehensive individual toxicity risk (CITOR) profile, comprising NTCP models for a wide range of acute as well as late radiation-induced toxicities, both physician- and patient-reported [69]. Tambas et al., reported on their first experience after the clinical implementation of model-based selection between 2018 and 2019 [70]. For definitive RT, with or without systemic treatment, 172 out of 227 referred patients were eligible for the model-based selection procedure, of which 80 patients (35%) eventually qualified for proton therapy. Patients who were selected for PT mainly had mucosal SCC above the level of the hyoid (nasopharynx and oropharynx), while only 12% of patients with SCC of the larynx qualified for PT. This is in line with the patient populations in Table 1, and with a previous study by Jakobi et al. [71], suggesting that the model-based approach is able to select those patients that other institutions would also treat with PT, however, on a more objective basis.

The model-based approach is currently based only on 3 validated NTCP-models for (1) patient-rated moderate to severe xerostomia [72], (2) physician-rated grade ≥2 dysphagia [73] and (3) tube feeding dependence [74]. Since RT for HNSCC may result in a much wider range of acute and late toxicities, one could argue that currently not all relevant factors are considered in the assessment of which patients would benefit from PT. Furthermore, all three endpoints were assessed at six months after the completion of RT. It could be argued that the models do not cover the full spectrum of acute and late toxicities, and that the long-term benefit of PT over photon therapy may be overestimated. To address this potential issue, all patients of the University of Groningen are currently included in a data registry program. This will not only allow researchers to progress to multivariable NTCP-models and CITOR profiles, but also to incorporate long-term toxicities into future models and assessments. Another minor disadvantage of the model-based approach is that dose uncertainties are known to have an impact on the accuracy of model-based selection [75]. Indeed, NTCP models assume a constant dose per fraction, as well as a spatially uniform dose within the organ. In reality, the biological effect might be underestimated, although Bortfeld et al., reported the impact to be negligible for inter-fraction variations under 10% [76]. Moreover, model-based selection is time-consuming and requires considerable resources, although, in the Netherlands, this problem was apparently quickly rectified after a short learning period. Tambas et al., reported the median time between the first consultation and the first fraction of RT to be 16 days and 14 days in patients who were, and were not, selected for plan comparison, respectively [70]. Additionally, resources for plan comparison will be further reduced by the implementation of automated treatment planning procedures. Kouwenberg et al., investigated the potential of automated planning in combination with machine learning for the preselection of HNC patients, reaching an accuracy of 87% [77]. Recently, a decision support tool was developed to select HNC patients for PT before the start of RT planning, predicting the toxicity reduction with PT using only delineation data. The positive predictive value of the tool exceeded 90% [78]. Such tools will contribute to a more effective and time-efficient selection of HNSCC patients at a much earlier stage, reducing possible treatment delay.

Model-based selection seems the most objective approach to select patients for PT. We should, however, keep in mind that the approach assumes that the target dose remains biologically equivalent, and that tumor control probability (TCP) is not affected. Furthermore, a critical factor of the model-based selection method is the quality of the treatment plans under evaluation, enhancing the importance of gained experience in PT planning. In RT planning, it can be decided to spare specific OAR at the cost of others, which renders the gain in NTCP observer-dependent.

## 4. Technical Limitations

As the exact location of the Bragg peak and the subsequent sharp distal dose fall-off are uncertain in PT, the technique is more sensitive to the varying densities it travels through [79,80]. Multiple factors can shift the Bragg peak location, such as positioning errors, artifacts, tissue deformations, and anatomic changes, which are common in HNC patients due to either weight loss or tumor response [81,82,83]. A short and reliable beam path needs to be foreseen to mitigate the impact of these uncertainties. Intensity-modulated proton therapy (IMPT), the most advanced form of PT, utilizes pencil-beam scanning, consisting of two pairs of scanning magnets that guide the beam to varying directions and depth. The modulation of the proton beamline allows for more precise coverage of irregular targets. Additionally, robust optimization techniques have been developed that account for positioning and range errors explicitly through the optimization of the treatment plan over a range of simulated error scenarios [84]. In IMPT planning, single-field (i.e., each proton beam separately covers the target volume) or multiple-field (i.e., the proton beams collectively cover the target volume) optimization can be applied. Robustly optimized IMPT plans have been shown to provide both superior target coverage robustness and OAR sparing, compared to margin-based plans, when evaluated for these considered uncertainties [85]. This, however, does not account for anatomic changes over the treatment course. Since changes in the patient’s anatomy during treatment have been established as the most significant source of range uncertainty [86], several institutions frequently monitor the patient’s anatomy with regular imaging during the course of the treatment to trigger adaptation if necessary [87,88]. Given the wide range of variability in anatomic changes of TV and OAR in HNC patients, it has been, to this day, unclear for which patients treatment adaptation will be necessary. Furthermore, offline plan adaptation is particularly labor-intensive and, therefore, represents a significant personnel cost. Moreover, as a few days are typically needed for adaptive planning before proceeding to the delivery of the new adapted treatment plan, daily inter-fractional changes, e.g., variation of the nasal cavity-filling, are not adequately considered. There has been growing interest in clinically implementing an online daily adaptive proton therapy (DAPT) workflow to optimally manage inter-fractional changes [89,90]. In an interesting review by Albertini et al., the different steps that need to be taken before the introduction of DAPT are discussed [86]. One of the main bottlenecks of DAPT is time-efficiency and the accuracy of the contouring process on the daily image. It is believed that artificial intelligence will play an essential role in this matter, although more research in understanding the clinical impact of uncertainties in volume definition is needed. In addition to online adaptation, the added value and feasibility of multiple-CT robust optimization is currently also being investigated [91,92,93].

Very recently, Scandurra et al., performed a longitudinal evaluation of plan robustness over the treatment course in 25 patients with NPC. Deformable image registration was used on weekly repeat CTs (rCT) to accumulate the nominal minimum and maximum rCT dose distribution and to investigate changes to target coverage and normal tissue dose. Only two patients required a plan adaption due to reduced target coverage. Both patients had advanced nodal spread (N2-disease) with suboptimal target coverage in the nodal neck region. Maximum doses to the critical OARs remained acceptable in all 25 patients, with some variations in individual cases. Significant weight loss appeared to have the largest impact on grade 2 or higher xerostomia NTCP. The authors conclude that robustly optimized IMPT plans, in combination with volumetric verification imaging and adaptive planning, provide adequate target coverage and acceptable OAR dose variation [94]. The results of this series might not be generally translatable, given the fact that NPC patients might experience less weight loss compared to other HNC locations. Nevertheless, Jiří et al., reported that nearly all patients with NPC in their cohort required adaptations due to a >5% change in dose to the target volume or OAR [33]. For OPSCC, robustly optimized IMPT has been investigated in the post-operative setting by Hague et al., who reported that none of the six patients required adaptation [95]. Gunn et al., found 38% of OPSCC patients to require adaptive re-planning because of weight loss and tumor volume changes. Re-scanning was performed at weeks 1 and 4, or on a case-by-case analysis. Yang et al., reported that 40% of all HNC patients treated with IMPT in their center required at least one plan adaptation [92]. Other in-silico, small cohort studies have reported inadequate target coverage in 25–60% of cases after robustly optimized IMPT planning [93,96]. Several small series have reported that the use of CBCT with deformable image registration does not suffice for accurate plan adaptation [97].

These results confirm that it remains unclear and rather unpredictable which patients may require plan adaptation, and at which time point it should occur during their treatment. The technical limitations that are faced in PT for HNSCC should be taken seriously, as they account for approximately 20% of the listed reasons for not treating patients with PT in Europe [18]. Software tools need to become available, not only for the implementation of adaptive radiotherapy, but also to facilitate remote treatment plan exchange between photon and PT centers for efficient plan comparison. A first step to this end was taken by Lühr et al., with the development of the web-based software tool ReCompare (REmote COMparison of PARticlE and photon treatment plans) [97].

## 5. The Issue of Clinical Trials and Reimbursement

Given the lack of level I evidence of PT for patients with HNSCC, IMRT currently remains the standard of care treatment [98]. Healthcare authorities from several countries require high level evidence documenting the clinical benefit of PT from randomized controlled trials (RCTs) before considering reimbursement of PT. This is understandable, since the evidence to date, including the non-randomized studies summarized in Table 1, have inherent selection bias. There is, however, an ongoing discussion about the value of phase III RCTs for the evaluation of new RT technologies. Some have argued that, when using the assessment paradigm used for drug approval, high-level evidence is rarely achievable [99]. With a new technology like PT, RCTs are indeed more challenging owing to technological complexity. First of all, multiple series proved that the quality of RT treatment planning and delivery is highly dependent on patient volume and varies widely among institutions, for both IMRT [100]and PT [101]. This finding renders the comparison between IMRT and PT difficult, since there is neither a standard PT nor a standard IMRT. Secondly, considering the rapid developments in both photon and PT planning, there is a possibility that at the time results of RCTs become available, their outcome will be based on or compared with outdated technology. Therefore, in the past, new RT techniques have rarely been introduced into clinical practice on the basis of the results of RCTs. Similar demands were not required for the implementation of IMRT, which was accepted as superior to three-dimensional conformal radiation therapy (3D-CRT) despite the increased cost [5]. There is also the issue of a patient’s willingness to be randomized when informed that one alternative results in unnecessarily irradiating normal tissues that can be avoided with the alternative. Furthermore, some authors argue that even large, highly powered, and well-founded RCTs probably will not provide the desired information. In the case of a “positive” trial, a large proportion of HNSCC patients will meet the eligibility criteria of the patients included in the trial and will receive PT, even though they might not experience any significant clinical benefit. Conversely, owing to the dilution of any effect by inclusion of an even larger proportion of patients who predictably will not benefit, a “negative” RCT might prohibit treating patients with PT, including the ones that may benefit [102]. The value of RCTs also depends on the endpoints that are prioritized. Most European PT institutions indicate that patients are selected for PT with the primary aim to decrease the risk of radiation-induced side effects, with an equivalent target dose [18]. If endpoints were local control or toxicities such as RD, RCTs would be more appropriate. In an interesting paper, Langendijk et al., discusses alternative evidence-based approaches for clinical validation PT [102]. The authors propose either the model-based approach with continuous validation, as explained earlier, or the so-called cohort multiple RCTs, in which a large, specific cohort of patients is followed prospectively after PT [103].

Lack of reimbursement is an important reason for not treating eligible HNSCC patients with PT [104,105]. Conversely, phase II and III trials with PT are rare due to reimbursement and funding issues [18]. We are, therefore, facing a self-sustaining problem. A retrospective analysis of 444 patients in the USA showed that insurers have not implemented ASTRO’s model policy, ultimately denying treatment for more than one third of the patients eligible for PT [106].

A major variability thus exists among countries, underlining the need for international guidelines and collaboration between stakeholders on reimbursement [70]. A compromise may be found in the approach of *coverage with evidence development*, in which evidence is collected in an ongoing manner in population-based registries along with dedicated financing [107].

## 6. Costs-Effectiveness of Proton Therapy

Taking into account the increased cost compared to photon treatment and the limited evidence for PT, the cost-effectiveness of PT is heavily debated [108,109]. Based on early clinical experience, the use of PT will likely result in a clinically apparent reduction of grade 2 or higher acute and subacute toxicities. As grade 2 or higher AEs are proven to affect the patient’s QOL and often require medical intervention, the implementation of PT may potentially result in a reduction in subsequent healthcare resources [110]. However, the question remains as to what extent do the costs of PT compared to photon therapy translate into a relevant reduction of healthcare costs. Several comprehensive cost-effectiveness analyses calculating quality-adjusted life years (QALYs) have been performed. Brodin et al., found PT to be cost-effective in more than 50% of patients with p16-positive OPC tumors treated with comprehensive nodal irradiation [111]. Cheng et al., found that IMPT would be cost-effective at €80,000 per spared QALY for 8 of 23 HNC patients [64], while Ramaekers et al., reported PT to not be cost-effective for HNC patients compared to IMRT [68]. Sher et al., performed a cost-effectiveness study by developing a Markov model to compare PT and IMRT for a 65-year-old patient with stage IVA OPSCC. The authors conclude that IMPT is never cost-effective in the societal perspective, while it may be cost-effective in the payer’s perspective only in younger, HPV-positive patients when profound improvement in dysphagia and xerostomia are assumed [112].

It should be considered that both the upfront treatment cost and the costs related to managing complications may vary considerably between different countries and institutions, which could explain the varying results. The reported series suggest that PT is most beneficial for patients with OPSCC with high risk of experiencing treatment-related normal tissue complications, together with longer life expectancy. Importantly, though, for sub-groups of patients such as those with p16-negative OPSCC treated with unilateral neck irradiation, PT may not be cost-effective even at a lower cost, as the estimated QOL benefit is minimal for some of these patients. A limitation of cost-effectiveness analyses is that not all possible treatment complications are considered, leading to a possible underestimation of the cost-effectiveness of PT. In addition, there is still insufficient knowledge about the reduction of late radiation-induced side effects and secondary tumor induction, which is particularly relevant for young patients with good prognosis. Furthermore, uncertainties in the assumptions underlying the cost-effectiveness analyses can limit interpretation and applicability [113]. The limitations of cost-effectiveness analyses can be countered by using time-driven activity-based costing (TDABC), a bottom-up cost accounting method that measures the cost of resources used based on the actual time that personnel and equipment are used to treat patients. In a recent case-matched pilot study of 50 patients with newly diagnosed oropharyngeal (OPC) cancer, the costs of delivering IMPT and photon therapy were compared [114]. Although IMPT was on average more costly than IMRT, primarily owing to higher equipment costs, a subset of IMRT patients had similar costs to IMPT patients, owing to greater use of supportive care resources. The authors conclude that the TDABC approach might be useful for the comparison of IMPT and IMRT, not only over entire care cycles, but also at the per-patient level to identify subsets of patients in which higher upfront costs may ultimately be of higher value.

What we should conclude, above all, is that a significant variability exists in the estimated cost-effectiveness between individual patients. This supports the individualized model-based approach for the decision whether to offer PT to an individual patient.

## 7. Future Directions

Encouraged by the early clinical experience, several randomized trials have been launched to investigate the effectiveness of PT in terms of oncologic outcomes and toxicity reduction [17]. Some series limit inclusion criteria to one HNC subsite, e.g., NCT01893307 and NCT04609280 including locally-advanced OPSCC, and NCT04528394, NCT00592501 and NCT02135042 including only NPSCC [115]. Other trials include multiple HNSCC locations, e.g., NCT03513042 and NCT04870840. The DAHANCA 35 trial is open for SCC of both the oropharynx and the larynx. Furthermore, several trials focus on patients with previously irradiated HNC (NCT01973179, NCT03217188, NCT03164460, NCT04671667, NCT03539198 and DAHANCA 37). In the period 2022-2025, the construction of another 25 new PT centers with 67 therapy rooms is foreseen [116].

## 8. Conclusions

The evidence summarized in this review suggests that PT is a promising treatment option for patients with HNSCC, particularly for those with mucosal SCC above the level of the hyoid. Further prospective well-designed investigations comparing the clinical effectiveness and benefits of proton beam therapy to IMRT are necessary, especially with regard to long-term toxicity. Radiation treatment plan optimization methods to account for inter-fractional anatomical changes should be improved. To enable wider implementation of PT to patients with HNSCC, radiation oncologists, health economists, patient advocates, and governments should critically reflect on how to evaluate reimbursement for HNSCC patients. Model-based selection with continuous validation currently represents the most evidence-based approach to select individual patients for PT for maximal cost effectiveness.

## Figures and Tables

**Figure 1 cancers-14-02587-f001:**
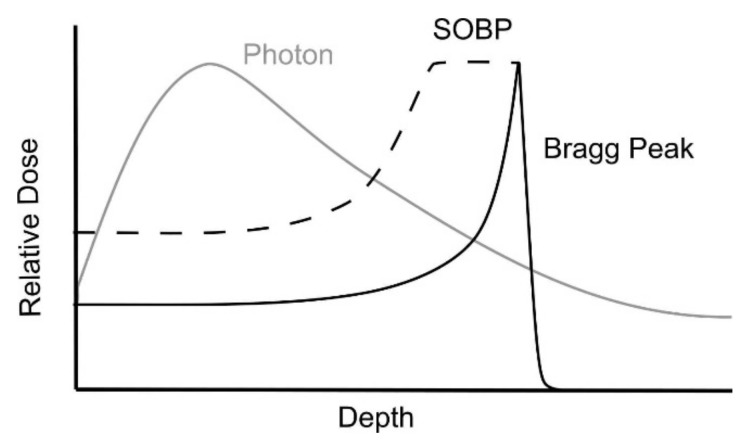
Depth-dose curves for photon and proton beams. Photons, with no mass and no charge, are highly penetrating, delivering dose throughout any volume of tissue. By contrast, heavy and charged proton particles deposit their energy mostly at the end of the particle path and, thus, deeper in the tissue. This phenomenon is reflected on the depth dose distribution, which has a typical peak (Bragg’s) followed by sharp dose fall-off. Modulated proton beams allow for spread-out Bragg peak (SOBP), covering the tumor target at various depths, at the cost of an increased entrance dose.

**Table 1 cancers-14-02587-t001:** Study characteristics, toxicity, and oncologic outcomes of all series reporting experience with single-modality PT for HNSCC, ordered by year published. Studies investigating other heavy ion particles e.g., carbon, non-SCC etiologies, and multiple HNC locations were excluded.

	TUMOR SITE	STUDY TYPE	NUMBER OF PATIENTS	MEDIAN FOLLOW-UP (IMRT-PT)	TOXICITY IMRT VS. PT	ONCOLOGIC OUTCOME IMRT VS. PT
**Holliday et al., (2015)** [25]	NPC	RETROSPECTIVE CASE-MATCHED COHORT	10 IMPT 20 IMRT	21.6 MO 25.8 MO	**ACUTE G3** **TOXICITY:** 90% vs. 50% * **LATE G2** **DYSPHAGIA:** 15% vs. 0% **ACUTE G3** **DERMATITIS:** 25% vs. 40% **G-TUBE:** 65% vs. 20% *	*AT LATEST FUP:* **LRC:** 95 vs. 100% **OS:** 95% vs. 90%
**Sio et al., (2016)** [26]	OPC	RETROSPECTIVE	35 IMPT 46 IMRT	7.7 MO 2.7 MO	**TOP 11 MD** **ANDERSON SYMPTOMS:** NO DIFFERENCE IN ACUTE AND CHRONIC SYMPTOMS SIGNIFICANT DIFFERENCE IN FAVOR OF PT FOR SUBACUTE APPETITE AND MUCOSITIS * **G-TUBE**: 48% vs. 20% *	-
**Gunn et al., (2016)** [27]	OPC	PROSPECTIVE	50 IMPT	29 MO	**ACUTE G3** **MUCOSITIS:** 46% **ACUTE AND LATE G3** **DYSPHAGIA:** 24% and 6.12% **G-TUBE:** 22%	**2Y LRC:** 92% **2Y PFS**: 88.6% **2Y OS:** 94.5%
**Blanchard et al., (2016)** [28]	OPC	RETROSPECTIVE CASE-MATCHED COHORT	50 IMPT 100 IMRT	33 MO 29 MO	**ACUTE** **XEROSTOMIA GRADE 2–3:** 60% vs. 42%* **G-TUBE OF WEIGHT LOSS > 20% 3MONTHS AFTER RT**: 34% vs. 18%*	**3Y LRC:** 89.7% vs. 91% **3Y PFS:** 86.4% vs. 85.8% **3Y OS:** 89.3% vs. 94.3%
**Lewis et al., (2016)** [16]	NPC	PROSPECTIVE	10 IMPT	24.5 MO	**ACUTE G3** **DERMATITIS:** 44% **LATE G2** **XEROSTOMIA:** 22% **LATE ≥ G3** **TOXICITIES:** 0%	**2Y LRC**: 100% **2Y OS**: 88.9%
**Zhang et al., (2017)** [29]	OPC	RETROSPECTIVE	50 IMPT 534 IMRT	33.8 MO	**MANDIBULAR ORN:** 7.7% vs. 2% MEAN MANDIBULAR DOSE: 41.2% vs. 25.6%*	-
**Aljabab et al., (2020)** [30]	OPC	RETROSPECTIVE	46 IMPT (28 DEFINITIVE RT, 18 PORT)	19.2 MO	**G3 DERMATITIS:** 76% **LATE G2** **DYSPHAGIA:** 2% **LATE G2** **XEROSTOMIA:** 30%	*AT LATEST FUP:* **LRC:** 100% **PFS:** 93.5% **OS:** 95.7%
**Manzar et al., (2020)** [31]	OPC	RETROSPECTIVE	46 IMPT 259 IMRT (138 DEFINITIVE RT, 167 PORT)	30 MO 12 MO	**G-TUBE:** 56% vs. 25% * **ACUTE G3** **DYSPHAGIA:** 44.2% vs. 23.3% **ACUTE ≥ G2** **DERMATITIS:** 68.3% vs. 82.9% *	**1Y OS:** 91.3% vs. 92.6%
**Kitpanit et al., (2020)** [32]	OPC	RETROSPECTIVE	27 IMPT (18 DEFINITIVE RT, 9 PORT)	19 MO	**ACUTE G1-2** **DERMATITIS:** 92.6% **ACUTE G1-2** **DYSPHAGIA:** 81.5% **ACUTE G3** **TOXICITIES:** 3.7% **LATE G1** **XEROSTOMIA:** 77.8%	**1Y LRC:** 100% **1Y OS:** 100%
**Jiří et al., (2021)** [33]	NPC	RETROSPECTIVE	40 IMPT	24 MO	**ACUTE G3** **DERMATITIS:** 14% **G-TUBE:** 9.3% G2 **XEROSTOMIA:** 7% **G2 DYSPHAGIA:** 5%	**2Y LRC**: 84% **2Y PFS**: 75% **2Y OS**: 80%
**Li et al., (2021)** [34]	NPC	RETROSPECTIVE CASE-MATCHED COHORT	28 IMPT 49 IMRT	37 MO 23 MO	**ACUTE ≥ G2** **TOXICITIES:** 93.9% vs. 69.9% * **LATE G3** **TOXICITIES:** 16.3% vs. 3.8%	**2Y LRC:** 86.2% vs. 100% **2Y PFS:** 76.7% vs. 95.7% **3Y OS:** 94% vs. 100%
**Williams et al., (2021)** [35]	NPC	RETROSPECTIVE	26 IMPT	25 MO	**ACUTE G3** **DERMATITIS:** 42% **LATE G2** **XEROSTOMIA:** 8% **LATE G2** **DYSPHAGIA:** 4% **NO G3** **TOXICITIES**	**2Y LRC:** 92% **2Y OS:** 85%
**Cao et al., (2021)** [36]	OPC	RETROSPECTIVE	103 IMPT 429 IMRT	36.2 MO	**G2-3** **XEROSTOMIA:** − **< 18 MO:** 16% vs. 9% −**18–24 MO:** 20% vs. 6% * −**24–36 MO:** 20% vs. 6% *	**-**
**Chou et al., (2021)** [37]	NPC	RETROSPECTIVE CASE-MATCHED COHORT	80 IMPT 80 IMRT	24.1 MO 42.2 MO	**G-TUBE:** 15% vs. 5%* **WEIGHT LOSS > 7%:** 6.21 vs. 4.87 * **G3 DERMATITIS:** 7.5% vs. 35%* **NO** **DIFFERENCES FOR OTHER ACUTE ≥ G2** **TOXICITIES**	**2Y LRC:** 95% vs. 97.5% **2Y PFS:** 83.7% vs. 94.4% **2Y OS:** 89.5% vs. 100%

Abbreviations: NPC, nasopharyngeal carcinoma; OPC, oropharyngeal carcinoma; IMRT, intensity-modulated radiotherapy; IMPT, intensity-modulated proton therapy; G-tube, gastrostomy tube; LRC, loco-regional control; PFS, progression-free survival; OS, overall survival, G1, grade 1; G2, grade 2; G3, grade 3; ORN, osteoradionecrosis; PORT, postoperative RT; FUP, follow-up; MO: months; * Statistically significant difference with *p* < 0.05.

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
