# Peer review of "Proton Therapy for Squamous Cell Carcinoma of the Head and Neck: Early Clinical Experience and Current Challenges†"

_cancers, 2022, doi:10.3390/cancers14112587_

Round 1

Reviewer 1 Report

This manuscript is a very well written and very comprehensive overview of the literature concerning the use of protons for head and neck cancer. All important subjects are discussed and the arguments are well weighted. Indeed I agree that for some head and neck cases protons will be better than photons, so we need validated criteria the select the proper cases. As stated very clear by the authors most advantage is seen for acute toxicity. However, late toxicity may be most important for the patients. We need to the incorporate patients’ association to hear from them wat they find most important. One main issue is the cost: the authors mention 70-150%, but this is optimistic, including all costs (equipment, extra personnel, extra imaging, travel distance) it may be 250% and more.

One point of concern that is only briefly mentioned is the uncertainty about the RBE on the end of the Bragg peak. There is a possibility that the toxicity, not at the OAR, but in the GTV/ CTV region may be higher and the risk of a second tumor is higher (most second tumors arise in the high dose region), see:

Krishna Luitel et all, Proton radiation-induced cancer progression; Life sciences in space research, 19 (2018) 31 – 42.

Manem ett all: the effect of radiation quality on the risks of second malignancies, IJRbiology March 2015; 91 (3): 209-2017

Peter Inskipet all  Radiation related new primary solid cancers in the childhood cancer survivor study: comparative radiation dose response and modification of treatment effects. IJROBPh vol 94 number 4: 2016: 800-807:

Berrington de Gonzalez et all Second solid cancers after radiation therapy: a systematic review of the epidemiologic studies of the radiation response relelationship: IJROBPh 2013 number 2: 224- 233

The problem of inducing a second tumor will only be relevant for young patients  (<40), never smoked, with HPV+ oropharyngeal cancer.

Author Response

We thank the reviewer for the positive appreciation of our work. 

  1. We need to the incorporate patients’ association to hear from them wat they find most important. One main issue is the cost: the authors mention 70-150%, but this is optimistic, including all costs (equipment, extra personnel, extra imaging, travel distance) it may be 250% and more.

As mentioned in the paper in paragraph 6, the high cost of protontherapy needs to be taken into account. The cost we refer to in our paper is based on published studies. The real cost will indeed be very much different between centers, depending on different policies in different countries, cost models used and so on. The 70-150% is based on published studies.

  1. One point of concern that is only briefly mentioned is the uncertainty about the RBE on the end of the Bragg peak. There is a possibility that the toxicity, not at the OAR, but in the GTV/ CTV region may be higher and the risk of a second tumor is higher (most second tumors arise in the high dose region), see:

Krishna Luitel et all, Proton radiation-induced cancer progression; Life sciences in space research, 19 (2018) 31 – 42.

Manem ett all: the effect of radiation quality on the risks of second malignancies, IJRbiology March 2015; 91 (3): 209-2017

Peter Inskipet all  Radiation related new primary solid cancers in the childhood cancer survivor study: comparative radiation dose response and modification of treatment effects. IJROBPh vol 94 number 4: 2016: 800-807:

Berrington de Gonzalez et all Second solid cancers after radiation therapy: a systematic review of the epidemiologic studies of the radiation response relelationship: IJROBPh 2013 number 2: 224- 233

The problem of inducing a second tumor will only be relevant for young patients  (<40), never smoked, with HPV+ oropharyngeal cancer.

We fully agree with the reviewer that the issue of secondary cancers is relevant in this matter.

We included a paragraph on this matter in lines 234-240:

‘Another factor regarding the short median follow-up time is the uncertainty around the estimations for the risk of secondary tumors. Due to remaining uncertainty about the RBE at the end of the Bragg’s peak, the radiation dose to the TV might be underestimated. High-dose irradiation has been proven to be correlated with increased risk of secondary tumor induction, with a linear dose-response curve for most organs[57,58]. In an in vivo study on mice, PT was found to cause more complex DNA damage compared to X-rays, leading to increased oxidative stress[59].’

In line 486 we incorporated a sentence to emphasize that the problem of secondary tumors is most relevant for the young patients: ‘.....which is particularly relevant for young patients with good prognosis’

Reviewer 2 Report

Well written review. Summarizing well the current evidence and problems of proton beam therapy for head and neck cancer. Relevant as reimbursing is indeed a challenge in proton beam therapy.

Author Response

We thank the reviewer for the positive appreciation of our work.

Reviewer 3 Report

This is thoughtful, careful overview which is a pleasure to read. I especially appreciate the discussion on persistence lack of level I evidence for PT effect. I have only minor comments:

  1. Comprehensibility of the text is slightly compromised due to abundant use of abbreviations. I recommend reducing their use.
  2. I recommend rephrasing a sentence on lines 57 and 64: the energy of charged particle is deposited everywhere in the tissue it crosses, but most of the energy is deposited at the end of the particle path – it means deeper in the tissue. This phenomenon is reflected on the depth dose distributions (called Bragg curve), which has typical peak (Bragg´s) followed by sharp fall-off.
  3. On the lack of laryngeal SCC studies - https://www.redjournal.org/article/S0360-3016(19)34126-4/fulltext#relatedArticles

Author Response

This is thoughtful, careful overview which is a pleasure to read. I especially appreciate the discussion on persistence lack of level I evidence for PT effect. I have only minor comments:

  1. Comprehensibility of the text is slightly compromised due to abundant use of abbreviations. I recommend reducing their use.

A thorough English editing was done by a native English speaking co-author, improving the readability of the paper

  1. I recommend rephrasing a sentence on lines 57 and 64: the energy of charged particle is deposited everywhere in the tissue it crosses, but most of the energy is deposited at the end of the particle path – it means deeper in the tissue. This phenomenon is reflected on the depth dose distributions (called Bragg curve), which has typical peak (Bragg´s) followed by sharp fall-off.

We added a paragraph on Bragg peak in lines 61-63:

‘By contrast, heavy and charged proton particles deposit their energy mostly at the end of the particle path and thus deeper in the tissue. This phenomenon is reflected on the depth dose distributions, which has a typical peak (Bragg’s) followed by sharp dose fall-off.’

  1. On the lack of laryngeal SCC studies - https://www.redjournal.org/article/S0360-3016(19)34126-4/fulltext#relatedArticles

We thank the reviewer for pointing us to this abstract and we included this study on laryngeal cancer in lines 158-160:

‘Only one abstract was published on the outcomes and toxicities of 14 patients treated with PT for laryngeal SCC. The authors report a high LRC and OS with a cumulative incidence rate of 12.3% and 8.3% for acute and late grade 3 toxicities, respectively [43].’